https://doi.org/10.1038/s41467-021-24272-9　　**OPEN**

# Anisotropic moiré optical transitions in twisted monolayer/bilayer phosphorene heterostructures

Shilong Zhao [1,2,10], Erqing Wang[1,10], Ebru Alime Üzer[3], Shuaifei Guo[4], Ruishi Qi[2,5], Junyang Tan[1], Kenji Watanabe [6], Takashi Taniguchi [6], Tom Nilges[3], Peng Gao [5,7], Yuanbo Zhang[4], Hui-Ming Cheng [1], Bilu Liu [1✉], Xiaolong Zou [1✉] & Feng Wang [2,8,9✉]

Moiré superlattices of van der Waals heterostructures provide a powerful way to engineer electronic structures of two-dimensional materials. Many novel quantum phenomena have emerged in graphene and transition metal dichalcogenide moiré systems. Twisted phosphorene offers another attractive system to explore moiré physics because phosphorene features an anisotropic rectangular lattice, different from isotropic hexagonal lattices previously reported. Here we report emerging anisotropic moiré optical transitions in twisted monolayer/bilayer phosphorenes. The optical resonances in phosphorene moiré superlattice depend sensitively on twist angle and are completely different from those in the constitute monolayer and bilayer phosphorene even for a twist angle as large as 19°. Our calculations reveal that the $\Gamma$-point direct bandgap and the rectangular lattice of phosphorene give rise to the remarkably strong moiré physics in large-twist-angle phosphorene heterostructures. This work highlights fresh opportunities to explore moiré physics in phosphorene and other van der Waals heterostructures with different lattice configurations.

[1] Tsinghua-Berkeley Shenzhen Institute, Tsinghua University, Shenzhen, China. [2] Department of Physics, University of California at Berkeley, Berkeley, CA, USA. [3] Department of Chemistry, Technical University of Munich, Garching, Germany. [4] State Key Laboratory of Surface Physics and Department of Physics, Fudan University, Shanghai, China. [5] Electron Microscopy Laboratory and International Center for Quantum Materials, School of Physics, Peking University, Beijing, China. [6] National Institute for Materials Science, Tsukuba, Japan. [7] Collaborative Innovation Center of Quantum Matter, Beijing, China. [8] Material Science Division, Lawrence Berkeley National Laboratory, Berkeley, CA, USA. [9] Kavli Energy NanoSciences Institute at University of California Berkeley and Lawrence Berkeley National Laboratory, Berkeley, CA, USA. [10] These authors contributed equally: Shilong Zhao, Erqing Wang. ✉email: bilu.liu@sz.tsinghua.edu.cn; xlzou@sz.tsinghua.edu.cn; fengwang76@berkeley.edu

Stacked van der Waals heterostructures with a finite twist angle can generate a moiré superlattice, which is characterized by a periodic variation of the interlayer stacking order. Such moiré superlattices can dramatically modify the electronic band structures of two-dimensional (2D) van der Waals heterostructures and have given rise to many fascinating quantum phenomena. For example, correlated insulator states[1,2], superconductivity[3–5], magnetism[6,7], and Chern insulators[6,8] have been observed in "magic angle" twisted bilayer graphene and in ABC trilayer graphene/hexagonal boron nitride (hBN) moiré superlattices, while moiré excitons have been reported in small-twist-angle transition metal dichalcogenide (TMDC) heterostructures, e.g., WS$_2$/WSe$_2$[9], MoSe$_2$/WS$_2$[10], and MoSe$_2$/WSe$_2$ heterostructures[11,12]. All these materials (graphene, hBN, and TMDCs) belong to the 2D hexagonal structures with the electronic bandgap lying at the vertices (K and K′ points) of the Brillouin zone[13–17], and prominent moiré superlattice effects have been observed in small-twist-angle (θ < 2°) heterostructures[1,3,9–12,18]. It will be highly desirable to explore new moiré heterostructures with different lattice configurations that may exhibit strong moiré physics even for large twist angles.

In this work, we report the first experimental study of rectangular moiré superlattices of twisted monolayer/bilayer phosphorene heterostructures. Phosphorene features a puckered honeycomb structure that forms an anisotropic rectangular unit cell[19–21], and it has a direct bandgap lying at the Γ point in the first Brillouin zone[19,22–24]. In addition, few-layer phosphorene is known to exhibit unusually strong interlayer interactions, which leads to a dramatic change of the direct bandgap from 1.73 eV in monolayer phosphorene to 0.62 eV in tetralayer phosphorene[19,25]. Here we demonstrate the moiré potential completely changes the electronic band structure and gives rise to a new set of optical transitions in twisted monolayer/bilayer

phosphorene heterostructure even for twist angles larger than 19°. This behavior is in striking contrast to other moiré systems, where prominent moiré effects exist only for twist angles smaller than a few degrees. The emerging optical resonances in the phosphorene moiré heterostructure are linearly polarized, and the polarization axis is closer to but different from the armchair direction of the bilayer phosphorene. Our ab initio density functional theory (DFT) calculations show that the remarkably large moiré physics at large twist angle originates from the Γ-point direct bandgap as well as the strong and stacking-dependent interlayer electron hybridization in twisted phosphorene heterostructures. The theory also reveals the important role of the underlying electron Bloch wavefunction in the interlayer coupling, which results in a strongly hybridized conduction band but a negligibly coupled valence band in twisted phosphorene heterostructure. The calculated optical responses of the monolayer/bilayer moiré superlattice agree well with our experimental observations.

## Results

**Emerging optical transitions in twisted monolayer/bilayer phosphorenes.** Figure 1a illustrates the configuration of a twisted monolayer/bilayer phosphorene heterostructure encapsulated between thin hBN layers. Few-layer phosphorene samples were first mechanically exfoliated onto the surface of polydimethylsiloxane (PDMS) thin films, and then transferred to 285-nm thick silicon dioxide/silicon (SiO$_2$/Si) substrates[25,26] (see "Methods"). The layer number of phosphorene was determined from the optical image contrast and verified by photoluminescence (PL) measurements. Thin hBN layers were also mechanically exfoliated from their bulk crystals. We then sequentially assembled the top hBN, monolayer phosphorene,

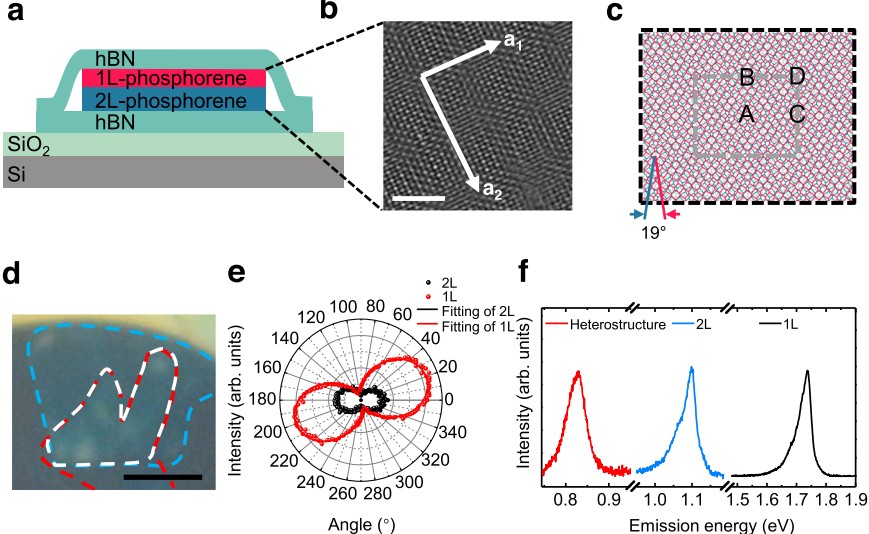

**Fig. 1 Representative twisted monolayer/bilayer phosphorene heterostructures. a** A side-view illustration of device's configuration, where 1L and 2L label monolayer and bilayer, respectively. **b**, HAADF-STEM image of a ~5.7° twisted monolayer/bilayer phosphorene heterostructure. It shows rectangular shape moiré superlattices, as indicated by two orthogonal superlattices vectors labeled **a$_1$** and **a$_2$**. Scale bar, 2 nm. **c** A top-view illustration of the twisted monolayer/bilayer phosphorene heterostructure with a large angle of 19°. The top monolayer (red) is rotated by 19° respected to the bilayer phosphorene (light cyan for the middle layer and blue for the bottom layer). The gray dashed rectangle indicates the supercell that contains four special stacking configurations, namely A, B, C, and D configurations. **d** The optical microscopy image of device D1. The red, blue, and white dashed lines indicate the monolayer, bilayer, and the twisted monolayer/bilayer phosphorene region, respectively. Scale bar, 5 μm. **e** Polarization-dependent PL emissions of the isolated monolayer and bilayer phosphorene, respectively. It shows a 19° twist angle between the top monolayer and bottom bilayer phosphorene. **f** PL spectra of the monolayer phosphorene (black), bilayer phosphorene (blue), and the 19° twisted phosphorene heterostructure (red). An emerging moiré optical transition at 0.83 eV is observed in the heterostructure, which is distinctly different from the monolayer resonance at 1.73 eV and the bilayer resonance at 1.10 eV. It shows that moiré superlattice strongly modulates optical transition in phosphorene heterostructure even for twist angles as large as 19°.

bilayer phosphorene, and the bottom hBN using a dry-transfer method[27,28] (see Methods). The whole structure was then transferred onto a 90-nm thick SiO$_2$/Si substrate for further optical measurements. To minimize sample degeneration, the whole fabrication process was done inside a nitrogen-gas-filled glovebox with both moisture and oxygen levels lower than 0.1 ppm.

To verify the formation of moiré superlattices in twisted monolayer/bilayer phosphorene heterostructures, we performed high-angle annular dark-field scanning transmission electron microscopy (HAADF-STEM) and high-resolution transmission electron microscopy (HRTEM) measurements on twisted monolayer/bilayer phosphorene heterostructures that were encapsulated by both top and bottom graphene (see "Methods"). Figure 1b shows an HAADF-STEM image of a ~5.7° twisted monolayer/bilayer phosphorene, which demonstrates clearly rectangular moiré superlattices, as indicated by two orthogonal superlattices vectors (white arrows). The rectangular moiré superlattice is further confirmed by the bright-field HRTEM images (indicated by blue arrows, Supplementary Fig. 1b) and its corresponding diffraction patterns (Supplementary Fig. 1c) of a ~4.5° twisted monolayer/bilayer phosphorene (Supplementary Fig. 1). Figure 1c illustrates the moiré superlattice of the phosphorene heterostructure with a large twist angle of 19°. The gray dashed rectangle indicates the supercell of the moiré superlattice with four special stacking configurations, namely, A, B, C, and D configurations (see Supplementary Fig. 4 for their atom configurations). Figure 1d shows the optical microscopy

image of a representative device (D1), where the monolayer and bilayer regions are outlined with red and blue dashed lines, respectively, and the twisted monolayer/bilayer phosphorene heterostructure exists in the overlapping region outlined by white dashed lines (see Supplementary Fig. 5 for details). We determined the twist angle between the monolayer and bilayer phosphorene through their anisotropic optical resonances. Figure 1e shows the polarization-dependent PL of the monolayer and bilayer phosphorene, from which we can determine a rotation of the principal axis of 19° ± 1° between the monolayer and bilayer. Two different devices (D2 and D3) with the monolayer–bilayer twist angle of 6° and 2° are shown in Supplementary Fig. 6. Figure 1f displays the PL spectra of the monolayer (black), bilayer (blue), and heterostructure regions (red) in the device D1, respectively. Surprisingly, we observe an emerging moiré optical transition at 0.83 eV in the twisted monolayer/bilayer phosphorene heterostructure, which is distinctly different from the monolayer resonance at 1.73 eV and the bilayer resonance at 1.10 eV. It shows that moiré superlattice has a dramatic effect on the optical properties of the phosphorene heterostructure even for twist angles as large as 19°.

Next we investigated optical resonances of phosphorene heterostructures with different monolayer–bilayer twist angles using photoluminescence excitation (PLE) spectroscopy. Figure 2a–d show the 2D color plot of the PLE results for the twisted phosphorene heterostructures with the twist angle at 19°, 6°, 2°, and 0° (i.e. exfoliated trilayer), respectively. The color scale bar corresponds to the PL intensities. In addition, the right panels

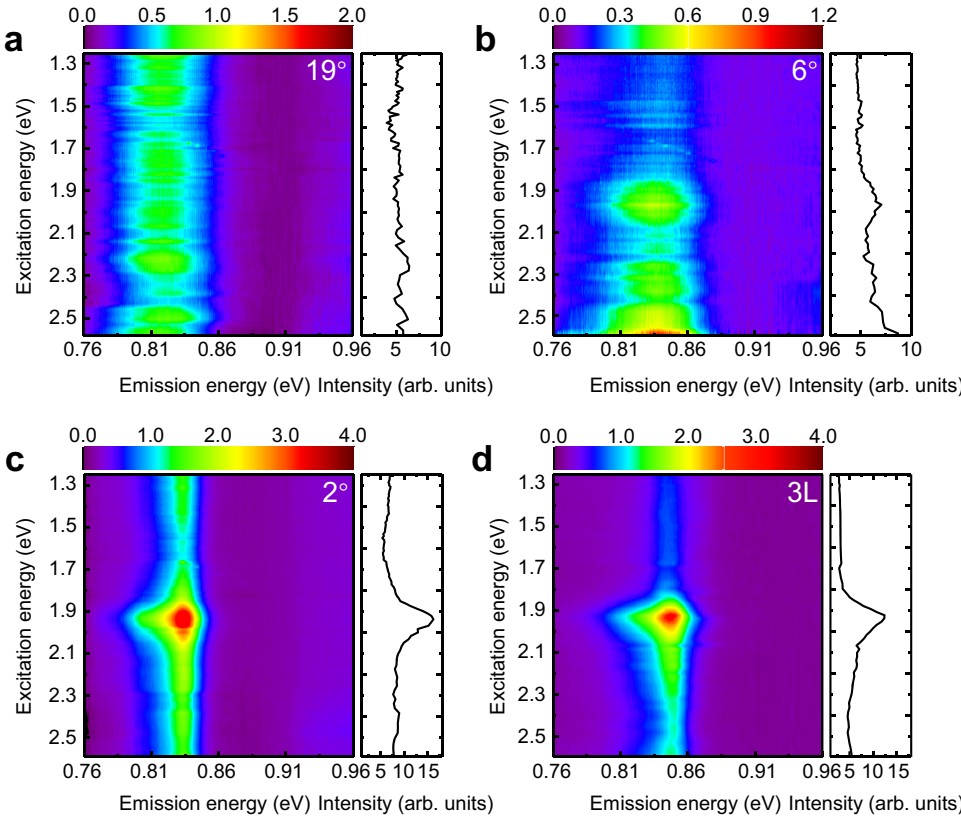

**Fig. 2 Twist-angle-dependent PLE spectra.** PLE spectra of twisted monolayer/bilayer phosphorene heterostructures with the angle of 19° (**a**), 6° (**b**), 2° (**c**), and a trilayer (labeled as 3L) phosphorene (**d**), respectively. The right panel in each figure shows the integrated PL intensity between the energy range of 0.8–0.9 eV. The top color scar indicates the PL emission intensities. All twisted phosphorene heterostructures show low energy PL emission close to 0.83 eV, but the high energy optical absorption spectra are completely different in different heterostructures: the 19° twisted heterostructure exhibits a broad absorption with no clear resonances, the 6° heterostructure has a weak absorption peak at 1.97 eV and a strong absorption peak at 2.64 eV, while the 2° heterostructure and trilayer phosphorene shows a single prominent absorption peak at 1.92 eV.

in the figure show the integrated PL intensity between 0.8–0.9 eV as a function of the excitation energy. The PLE spectra show that optical properties of the twisted monolayer/bilayer phosphorene heterostructures depend strongly on the twist angles, and they exhibit optical transitions distinctly different from the monolayer and bilayer phosphorene. All the twisted phosphorene heterostructures show low energy PL emission close to 0.83 eV, but there is a small energy shift among heterostructures of different twist angles (see Supplementary Fig. 7). In addition, the PL emission in large-twist-angle heterostructures (at 19° and 6°) is significantly broader. The high energy optical absorption in different heterostructures, however, are dramatically different: the 19° twisted heterostructure exhibits a broad absorption with no clear resonances, the 6° heterostructure has a weak absorption peak at 1.97 eV and a strong absorption peak at 2.64 eV, while the 2° heterostructure and trilayer phosphorene shows a single prominent absorption peak at 1.92 eV. The strongly twist-angle dependent optical transitions in monolayer/bilayer phosphorene heterostructures indicate that moiré potentials strongly modulate the electronic band structures of twisted monolayer/bilayer phosphorenes, and the moiré superlattice effect remains strong even in large-twist-angle heterostructures, which is different from reported systems with isotropic hexagonal lattice structures.

**Strong moiré effect in large-twist-angle phosphorene heterostructures**. We further investigated the polarization dependence of the PL emission in the 19° twisted phosphorene heterostructure (blue dots, Fig. 3a). The PL intensity shows a well-defined $\cos^2\theta$ pattern (blue line, Fig. 3a) when polarization angle $\theta$ varies, which indicates a linearly polarized emission from the twisted monolayer/bilayer phosphorene heterostructure. Interestingly, the polarization principal axis does not align with that of either the monolayer (red line) or the bilayer (black line). Instead, it is rotated by 4° (±1°) from the bilayer polarization principal axis. This behavior further demonstrates the strong modulation effect of moiré superlattice on optical resonances of the 19° twisted monolayer/bilayer phosphorene heterostructure.

To understand the unusual moiré superlattice effects on the electronic band structure and optical resonances of twisted phosphorene, we performed ab initio DFT calculations using the Vienna Ab initio Simulation Package (VASP[29], see "Methods"). We focused on the 19° heterostructure. Calculations on 6° and 2° twisted phosphorene heterostructure are beyond our computation capability due to their much larger unit cell sizes. We first benchmarked our DFT calculations in pristine monolayer, bilayer, and trilayer phosphorene, and our results reproduce

nicely the known optical transitions in these systems[19,30] (see Supplementary Fig. 10).

Figure 3b displays the calculated imaginary part of dielectric function (i.e., optical absorption spectrum) of 20.04° (referred as 20° for simplicity) twisted phosphorene heterostructure. The lowest optical resonance in the twisted phosphorene heterostructure is at 0.82 eV, which agrees well with the observed PL emission peak (black arrow in Fig. 3b). In addition, the spectrum shows a rather broad absorption between 1.3 and 2.6 eV with very weak resonances, consistent with the mostly featureless absorption spectrum observed experimentally for the 19° twisted heterostructure (black line, Fig. 3b). We also evaluated the angle-dependent optical absorption in the 20° twisted phosphorene heterostructure. Our calculation shows that the optical absorption is highly polarized in the heterostructure (see Supplementary Fig. 8). The polarization principal axis of the heterostructure lies between the polarization principal axis of the constituent monolayer and bilayer, and is rotated from the bilayer axis by ~7°. This is in qualitative agreement with the observed linear polarization of PL emission in the 19° twisted device.

Next we examined in detail the electronic structure of both the conduction and valence bands close to the Γ point in the twisted monolayer/bilayer phosphorene heterostructure, and compare it to the band structures of the monolayer, bilayer, and trilayer phosphorene. Figure 4a shows our calculation results for the monolayer (black solid line), bilayer (red solid line), trilayer (blue solid line), and 20° twisted heterostructure (orange solid line). The two horizontal dashed lines indicate the conduction band minimum (CBM) and the valence band maximum (VBM) of the bilayer phosphorene. It shows that the CBM decreases and the VBM increases progressively from monolayer to trilayer due to the interlayer coupling, as have been established in previous studies[19]. The band structure of the 20° twisted phosphorene heterostructure, however, exhibits a very different behavior. It has a VBM energy that is almost identical to that of the bilayer phosphorene, but a CBM energy much lower than those of both bilayer and trilayer phosphorene. The unusual electronic bands of the twisted phosphorene heterostructure are further illustrated in the partial charge density distribution of the CBM and VBM, as shown in Figs. 4c and d, respectively. Apparently, the conduction band is strongly hybridized between the monolayer and bilayer. Electrons at the CBM are mostly localized at the monolayer–bilayer interface, and the charge density exhibit strong periodic modulation (Fig. 4c). Electrons at the VBM, however, are mostly localized in the bottom bilayer with negligible monolayer–bilayer couplings (Fig. 4d). Our calculations also reveal that although the 20° twisted heterostructure and

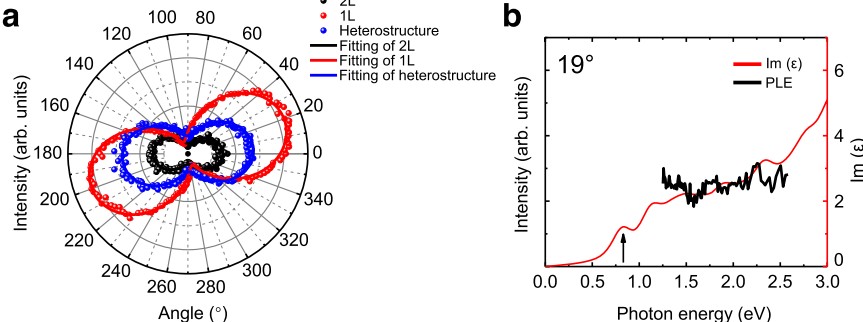

**Fig. 3 Optical properties of the 19° twisted phosphorene heterostructure. a** Polarization-dependent PL emissions (dots) and the corresponding fitting curves (solid lines) of the monolayer (red), bilayer (black), and the twisted phosphorene heterostructure (blue). The heterostructure shows a linearly polarized emission, and its principal axis is rotated by 4° from the bilayer polarization principal axis. **b** Comparison between the PLE spectra (black solid line) and the calculated imaginary parts of dielectric functions (red line) of the 19° twisted phosphorene heterostructure. The black arrow indicates the PL emission energy for the 19° twisted heterostructure.

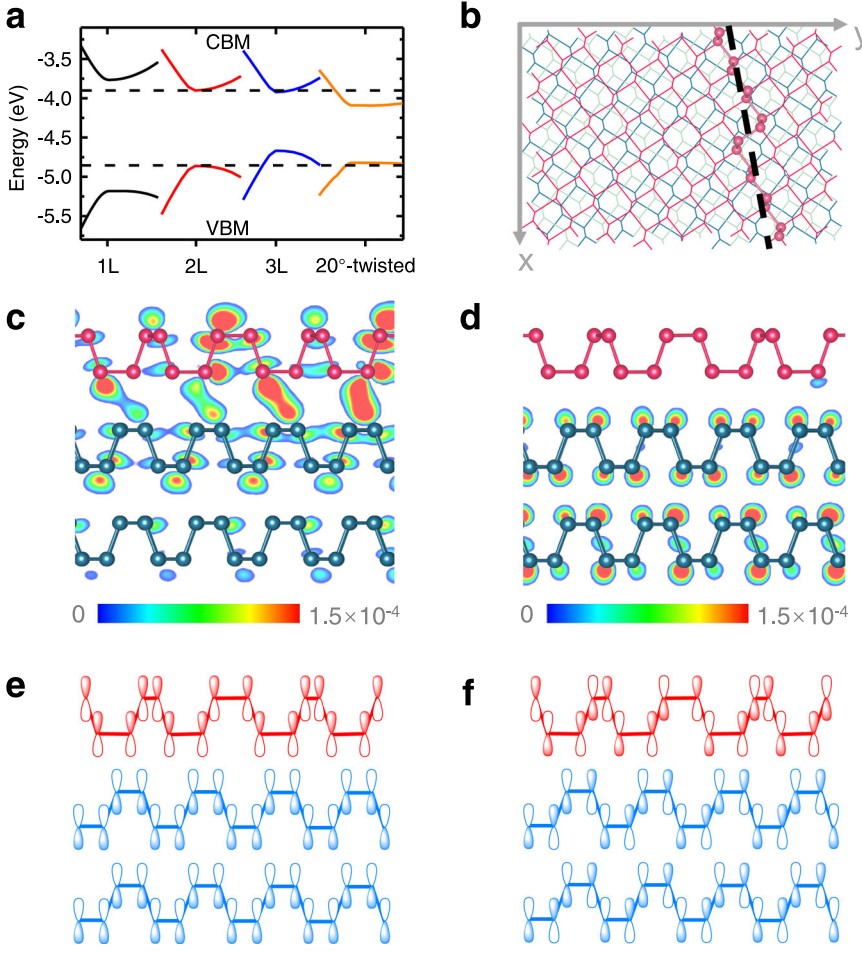

**Fig. 4 Strong interlayer coupling in the 20° twisted monolayer/bilayer phosphorene heterostructure. a** Band alignment of the monolayer (black solid line), bilayer (red solid line), trilayer (blue solid line), and 20° twisted phosphorene heterostructure (orange solid line) near Γ points. The two black dashed lines indicate the CBM and VBM of bilayer phosphorene. The vacuum energy is set to be at 0 eV. **b** The schematic top-view of the supercell of the 20° twisted phosphorene heterostructure. The red, cyan, and blue colors indicate the top, middle, and bottom phosphorene layers, respectively. The black dashed lines indicate the section plane for the partial charge density distributions, where the nearest phosphorus atoms are displayed in a ball-and-stick model. **c, d** The sectional partial charge density distribution at the CBM (**c**) and VBM (**d**) of the 20° twisted phosphorene heterostructure along the armchair direction of the constituent bilayer phosphorene, as indicated by the black dashed lines in (**b**). The isosurface levels are set to be $1.5 \times 10^{-4}$ e$\mathring{A}^{-3}$, as indicated by color scale bar. The wavefunctions of the monolayer (red) and bilayer phosphorene (blue) are strongly coupled at the CBM, whereas, they are mainly localized in the bottom bilayer at the VBM. **e, f** Schematic illustration of the overlapping of wavefunctions at the CBM (**e**) and VBM (**f**) in 20° twisted phosphorene heterostructure. The CBM electron wavefunctions have the same sign at the monolayer–bilayer interface (**e**), leading to constructive interference of interlayer coupling at different atom sites and strong hybridization between the monolayer and bilayer phosphorene. On the other hand, the VBM electron wavefunctions exhibit oscillating signs at the monolayer–bilayer interface (**f**), resulting in destructive interference of interlayer coupling at different atom sites and weak coupling of valence band states.

trilayer phosphorene have similar bandgaps, the underlying conduction and valence bands are completely different.

## Discussion

The very different CBM and VBM coupling between the monolayer and bilayer phosphorene in the twisted heterostructure originates from the different electron Bloch wavefunctions at the CBM and VBM, as illustrated schematically in Figs. 4e and f, respectively. The red and blue colors represent the wavefunctions from the top monolayer and bottom bilayer phosphorene, respectively. The sign of the electron wavefunctions is indicated by the filled or blank spindle. Here, we focus on the $p_z$ orbital components of the wavefunctions because of their dominant contribution to the wavefunctions at the CBM and VBM (see Supplementary Fig. 11). We find that the conduction band electron wavefunctions have the same sign at the monolayer–bilayer

interface. Therefore, interlayer coupling at different atom sites adds up constructively, resulting in strong hybridization between the monolayer and bilayer phosphorene at the CBM (Fig. 4e). In contrast, the electron wavefunctions at the VBM exhibit oscillating signs at the monolayer–bilayer interface (Fig. 4f). As a result, the electron coupling at different atom sites interferes destructively with each other, giving rise to a weak interlayer coupling. Consequently, the VBM electrons are mostly localized in the bilayer phosphorene with negligible hybridization with the monolayer (Fig. 4d).

In summary, our experimental observations reveal that moiré superlattices can strongly modulate the electronic band structures and optical transitions of the twisted monolayer/bilayer phosphorene. The moiré superlattice effect remains strong even for a very large twist angle, highlighting new moiré physics that can emerge in van der Waals heterostructures with rectangular lattices.

## Methods

**Fabrication of twisted phosphorene structures**. Bulk black phosphorus crystals were synthesized from red phosphorus with Sn and $SnI_4$ as transport agents using a chemical vapor transport method[31,32]. Briefly, red phosphorus (500 mg), Sn (20 mg), and $SnI_4$ (10 mg) were sealed in an evacuated ampoule and heated up to 650 °C. The temperature was subsequently cooled down to 500 °C in 10 h and held at 500 °C for 10 h, followed by cooling down to 150 °C at a rate of 15 °C per hour to grow crystals. Few-layer phosphorene flakes were first exfoliated onto the surface of polydimethylsiloxane (PDMS) thin film (Gel-Pak) and then transferred onto 285-nm $SiO_2$/Si substrates. The layer numbers of few-layer phosphorenes were estimated by measuring their optical image contrast[19], where adding each phosphorene layer will increase the contrast of the optical image by ~8%. Thin hBN flakes were also mechanically exfoliated onto 285-nm $SiO_2$/Si substrates. Polyethylene terephthalate (PET) stamps were employed to sequentially pick up the top hBN, monolayer phosphorene, bilayer phosphorene, bottom hBN at 60 °C. The final structures were then released onto a 90-nm $SiO_2$/Si substrate at 90–110 °C, followed by dissolving PET residues in dichloromethane for at least 12 h. To minimize the degradation of phosphorene flakes, the exfoliation, identification, and assembly were done within a day. Moreover, all fabrication processes were done inside a nitrogen-gas-filled glove box with both the oxygen and humidity levels less than 0.1 ppm. The whole structures were then put in a cryostat and pumped down to vacuum ($<1 \times 10^{-6}$ mbar) for optical measurements.

**Optical measurements**. Samples were kept at the liquid nitrogen temperature for all optical measurements. PL measurements were conducted using a lab-build micro-PL setup in reflection geometry with a long working distance near-infrared objective (50x, N.A. 0.42) and the spectrometer equipped with both silicon and InGaAs detectors. PLE measurements were performed using a super-continuum laser (SC-Pro, YSL) as the excitation source. Tunable excitation light with the linewidth less than 0.5 nm was spectrally picked up by a grating and filtered out by suitable filters. The spot size of the focused light was ~2 μm. The spectra were normalized to both integration time and incident power. For polarization-dependent PL measurements, linearly polarized light was set with a Glan-Thompson polarizer and rotated by a half-wave plate.

**TEM measurements**. For HRTEM measurements, monolayer/bilayer twisted phosphorene heterostructures were encapsulated by few-layer graphene to avoid sample degradation. PET stamps were employed to sequentially pick up the top graphene, monolayer phosphorene, bilayer phosphorene, and bottom graphene at 60 °C, as described above. The whole structure was transferred to a Quantifoil TEM grid with 1.2 μm hole (658-200-AU) at ~90 °C and the PET stamp was dissolved in dichloromethane for 12 h[9]. The HRTEM was performed on a Titan ETEM G2 microscope operated at 80 kV.

HAADF-STEM measurements were performed on a Nion U-HERMES200 microscope at 60 kV. The beam-convergence semi-angle was 35 mrad and the collection semi-angle range was 80–210 mrad. The acquired images were denoised by block-matching and 3D filtering (BM3D) algorithm to remove Poisson-Gaussian mixed noise. Low-frequency amorphous background was suppressed by an FFT filter.

**First-principles calculations**. The supercell of twisted monolayer/bilayer phosphorenes was constructed based on the coincidence site lattice theory[33]. The commensurate 20.04° twisted monolayer/bilayer phosphorene heterostructure was used to model the electronic band structure and optical properties of the experimental 19° one.

We performed DFT calculations using the projector augmented wave method[34] and the plane-wave basis as implemented in the VASP. The cutoff energy of the plane-wave basis was set to be 450 eV. The vacuum thickness was set to be 15 Å to avoid the interaction between adjacent layers. In addition, the supercells were fully optimized until the residual force per atom was less than 0.01 eVÅ$^{-1}$. For monolayer, bilayer, and trilayer phosphorene, Monkhorst-Pack $10 \times 8 \times 1$, $20 \times 16 \times 1$, and $50 \times 40 \times 1$ k-meshes were adopted to sample the first Brillouin zones, calculate electronic band structures, and obtain absorption spectra, respectively. For large-angle twisted phosphorene heterostructures, the k-meshes were $5 \times 3 \times 1$. The non-local vdW density functional (vdW-DF)[35] with the optB88 exchange functional (optB88-vdW)[36,37] was employed to describe vdW interactions among layers. The modified Becke Johnson (mBJ)[38,39] exchange-correlation functional was adopted to get more accurate bandgap estimations. The charge gradient used in the mBJ calculations was extracted from the bulk black phosphorus, which gives $c = 1.1592$.

## Data availability

The data that support the findings of this study are available from the corresponding author upon reasonable request.

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

## Acknowledgements
S.Z. thanks the financial support from Tsinghua-Berkeley Shenzhen Institute (TBSI), Tsinghua University. Partial of device fabrication was supported by the National Key R&D Program of China (no. 2018YFA0307200), the National Natural Science Foundation of China (nos. 51722206 and 51920105002), and the Bureau of Industry and Information Technology of Shenzhen for the "2017 Graphene Manufacturing Innovation Center Project" (no. 201901171523). The theoretical part was financially supported by the National Key R&D Program of China (grant no. 2017YFB0701600), the National Natural Science Foundation of China (11974197), Shenzhen Basic Research Projects (no. JCYJ20170407155608882), and Guangdong Innovative and Entrepreneurial Research Team Program (grant no. 2017ZT07C341). S.G. and Y.Z. acknowledge financial support from National Key Research Program of China (grant nos. 2016YFA0300703 and 2018YFA0305600), NSF of China (grant nos. U1732274, 11527805, and 11421404), Shanghai Municipal Science and Technology Commission (grant no. 18JC1410300), and Strategic Priority Research Program of Chinese Academy of Sciences (grant no. XDB30000000). The growth of black phosphorus crystals was supported by the Deutsche Forschungsgemeinschaft (DFG, German Research Foundation) under Germany's Excellence Strategy e-conversion cluster EXC 2089/1-390776260. The growth of hexagonal boron nitride crystals was supported by the Elemental Strategy Initiative conducted by the MEXT, Japan and the CREST (JPMJCR15F3), JST. We acknowledge Dr. Chaw-Keong Yong, M. Iqbal Bakti Utama, Dangqing Wang, Dr. Wenyu Zhao, Emma C. Regan, and Halleh B. Balch for their help on the optical measurements.

## Author contributions
F.W. conceived the project. F.W., X.Z., and B.L supervised the experimental and theoretical studies. S.Z. fabricated twisted phosphorene heterostructures and performed optical measurements. E.W. and X.Z. performed the ab initio DFT calculations. S.G. and Y. Z. helped with the fabrication of twisted phosphorene heterostructures. F.W., S.Z., and B.L. analyzed the experimental data. E.A.Ü. and T.N. grew black phosphorus crystals. K.W. and T.T. grew hexagonal boron nitride crystals. S.Z. and J.T. performed HRTEM measurements under H.-M.C.'s supervision. R.Q. and P.G. performed HAADF-STEM measurements. All authors discussed the results and wrote the manuscript.

## Competing interests
The authors declare no competing interests.
