## [Peer Review File · Nature Communications]

REVIEWER COMMENTS

Reviewer #1 (Remarks to the Author):

The reviewer appreciates the authors answered half of my questions/concerns on the TEM image of Moire pattern from 1-layer/2-layer phosphorene. The good-looking TEM image of Moire pattern strengthens this manuscript. At least, there is a Moire pattern. However, if graphene layers are used to encapsulate 1-layer/2-layer phosphorene, it raises the same question how the authors can exclude top graphene/1-layer phosphorene and/or 2-layer phosphorene/bottom graphene layer cannot form Moire pattern. This concern comes from a widely reported TMDs Moire patterns. It seems any high-quality 2D/2D interface has the potential to form a Moire pattern. More convincing result would be from experiments showing graphene/1-layer phosphorene, graphene/2-layer phosphorene doesn't have or have totally different Moire patterns from the one the authors reported here. A simple answer of weak coupling between graphene and phosphorene is not that convincing to this reviewer.

Reviewer #3 (Remarks to the Author):

The authors have well addressed all my concerns. I would recommend its publication. Thanks.

Reviewer #1 (Remarks to the Author):

The reviewer appreciates the authors answered half of my questions/concerns on the TEM image of Moire pattern from 1-layer/2-layer phosphorene. The good-looking TEM image of Moire pattern strengthens this manuscript. At least, there is a Moire pattern. However, if graphene layers are used to encapsulate 1-layer/2-layer phosphorene, it raises the same question how the authors can exclude top graphene/1-layer phosphorene and/or 2-layer phosphorene/bottom graphene layer cannot form Moire pattern. This concern comes from a widely reported TMDs Moire patterns. It seems any high-quality 2D/2D interface has the potential to form a Moire pattern. More convincing result would be from experiments showing graphene/1-layer phosphorene, graphene/2-layer phosphorene doesn't have or have totally different Moire patterns from the one the authors reported here. A simple answer of weak coupling between graphene and phosphorene is not that convincing to this reviewer.

Our reply: We thank the referee for his/her comments.

Following the referee's suggestion, we performed HAADF-STEM measurement on 2-layer phosphorene that was encapsulated by both top and bottom graphene, as shown in Fig. R1. It shows the atomic configuration of 2-layer phosphorene without the signal of graphene, due to the large intensity contrast between phosphorus and carbon atoms ($I_P/I_C \sim 6$) in HAADF-STEM images.

To visualize the pattern formed by graphene/phosphorene, we performed simulations on the atomic configuration of graphene/phosphorene heterostructure using the same method as described in the Supplementary Information (see the caption of Fig. S2b). Figure R2 shows the patterns formed by 1-layer graphene/2-layer phosphorene heterostructure with different twist angles. Because graphene and phosphorene have totally different lattices (hexagonal versus rectangular)

and large lattice mismatch, graphene/phosphorene heterostructures form one-dimensional (1D) stripe-like patterns with small periodicities, which are completely different from the rectangular-shape pattern of twisted phosphorene. Note that the simulations in Fig. R2 only considered the lattice configuration of graphene/phosphorene heterostructure but didn't consider their chemical information (atomic number of elements).

The above results confirm that the observed rectangular-shape moiré pattern in Fig. S2a of the Supplementary Information is formed by twisted 1-layer/2-layer phosphorene rather than the interface between graphene and phosphorene.

Fig. R1. The HAADF-STEM image of bilayer phosphorene encapsulated by both top and bottom graphene.

Inset, a structural illustration of bilayer phosphorene.

Fig. R2. Simulated atomic configurations of graphene/2-layer phosphorene heterostructure with different twist angles. The twist angle is defined as the angle difference between the zigzag directions of graphene and phosphorene. The 1D stripe-like pattern (indicated by blue arrows) is completely different from rectangular-shape patterns formed by twisted 1-layer/2-layer phosphorene. Scale bar, 2 nm.

Reviewer #3 (Remarks to the Author):

The authors have well addressed all my concerns. I would recommend its publication. Thanks.

Our reply: We thank the referee for his/her recommendation.

Reviewer #1 (Remarks to the Author):

The authors clarified all of the concerns from this reviewer. I recommend publishing this manuscript at Nature Communications.

Our reply: We thank the referee for his/her recommendation.

REVIEWERS' COMMENTS

Reviewer #1 (Remarks to the Author):

The authors clarified all of the concerns from this reviewer. I recommend publishing this manuscript at Nature Communications.